# Existential suffering as a motive for assisted suicide: Difficulties, acceptability, management and roles from the perspectives of Swiss professionals

**Marie-Estelle Gaignard**[1,2]*, **Sophie Pautex**[3], **Samia Hurst**[1]

**1** Institute for Ethics, History and the Humanities, Faculty of Medicine, University of Geneva, Geneva, Switzerland, **2** Department of Oncology, Geneva University Hospitals, Geneva, Switzerland, **3** Division of Palliative Medicine Department of Rehabilitation and Geriatrics, Geneva University Hospitals, Geneva, Switzerland

* marieestellegaignard@gmail.com

**Data Availability Statement:** All relevant data are within the paper, its Supporting Information file and our previous published article*. * Gaignard M-E, Hurst S. A qualitative study on existential

## Abstract

### Background

Existential suffering is often a part of the requests for assisted suicide (AS). Its definitions have gained in clarity recently and refer to a distress arising from an inner realization that life has lost its meaning. There is however a lack of consensus on how to manage existential suffering, especially in a country where AS is legal and little is known about the difficulties faced by professionals confronted with these situations.

### Objectives

To explore the perspectives of Swiss professionals involved in end-of-life care and AS on the management of existential suffering when it is part of AS requests, taking into account the question of roles, as well as on the difficulties they encounter along the way and their views on the acceptability of existential suffering as a motive for AS.

### Methods

A qualitative study based on face-to-face interviews was performed among twenty-five participants from the fields of palliative and primary care as well as from EXIT right-to-die organization. A semi-structured interview guide exploring four themes was used. Elements from the grounded theory approach were applied.

### Results

Almost all participants reported experiencing difficulties when facing existential suffering. Opinions regarding the acceptability of existential suffering in accessing AS were divided. Concerning its management, participants referred to the notion of being present, showing respect, seeking to understand the causes of suffering, helping give meaning, working together, psychological support, spiritual support, relieving physical symptoms and palliative sedation.

suffering and assisted suicide in Switzerland. BMC Med Ethics. 2019 May.

**Funding:** MEG conducted this work without funding as part of her Doctorate in Medicine. The Institute for Ethics, History, and the Humanities at the Geneva University Medical School provided access to library facilities. The Institute had no role in designing the study, nor in its execution, analyses, interpretation and in writing the manuscript.

**Competing interests:** The authors have declared that no competing interests exist.

**Abbreviations:** AS, assisted suicide; *CCER*, Commission Cantonale d'Ethique de la Recherche (Swiss Cantonal Ethics Committee); EAPC, European Association for Palliative Care; EV, *EXIT* Volunteer; *EXIT*, name of a swiss right-to-die organization; Pn, Participant number; PallCP, Palliative Care Providers; PrimCP, Primary Care Providers; SAMS, Swiss Academy of Medical Sciences.

## Conclusion

This study offers a unique opportunity to reflect on what are desirable responses to existential suffering when it is part of AS requests. Existential suffering is plural and implies a multiplicity of responses as well. These situations remain however difficult and controversial according to Swiss professionals. Clinicians' education should further address these issues and give professionals the tools to better take care of these people.

## Background

Existential suffering is often a part of the requests for assisted suicide (AS) [1–3]. The acceptability of existential suffering as a motive for AS is still a complex and controversial issue that is regularly debated in the medical literature [4]. According to recent definition, existential suffering refers to a distress arising from an inner realization that life has lost its meaning [5–9]. Everybody can have existential concerns at any time of life but there is evidence that for patients at the end of life, those can increase their wish to hasten death [10]. As reported in our previous paper, existential suffering was interpreted by Swiss professionals confronted to AS as, literally, "a life that wasn't worth living any longer" and/or "a life that did not make sense anymore" in the patient's view, and this for many different reasons [11]. We found that in most cases existential suffering consisted of different, and sometimes compounded, losses of the dimensions of life such as physical decline with the resulting feeling of loss of autonomy and loss of identity, but also loneliness and loss of social significance.

A certain clinical clarity on what existential suffering is being acquired lately which should hopefully improve the evaluation and management of this kind of suffering in the coming years. Several questions then arise: how should we concretely manage existential suffering? After having explored and proposed other alternatives, is AS an acceptable solution to existential suffering? For recall, Switzerland has one of the most liberal legislations on AS in the world. The law does not specify requirements in terms of suffering. According to the SAMS (Swiss Academy of Medical Sciences), which provides healthcare professionals with recommendations on ethics issues, the assisting person has to verify that five conditions are present, including that the symptoms of the patient's disease and/or their functional impairments are a source of "intolerable suffering" for them and that "medically indicated treatment options and other types of assistance and support have been sought and have provided ineffective or are rejected as unacceptable by the patient" [12]. Lack of consensus on what these approaches are in the case of existential suffering, both in cases where it is deemed "intolerable" by patients and in cases where it is not, and on how to manage this form of suffering, often leads to a feeling of helplessness in physicians and other health care professionals [13–16].

Other authors, mainly from the palliative care field, have already addressed this issue and have suggested the importance of being present for the patient, the use of psycho-existential approaches and the promotion of spiritual care [5, 17–22]. A 2008 review [10] reported on eight "existentially-informed interventions" addressing existential concerns that have the potential to positively impact a person's well-being and overall quality of life at the end of life. Dignity therapy is one of the studied interventions that has certainly gained in recognition recently [23]. Although promising, these approaches are still too rarely used systematically in Swiss hospitals. Another new and promising therapeutic approach is the use of classic serotonergic psychedelics in the management of patients with refractory depression and/or anxiety at the end of life [24–26]. Indeed, to date, no pharmacological approach has shown a

significant benefit in the management of existential suffering. Palliative sedation is also subject to controversies [2, 14]. The European Association for Palliative Care (EAPC) stated in 2009 that "palliative sedation may be considered for severe non-physical symptoms such as refractory depression, anxiety, demoralization or existential distress", after taking into account several considerations [1, 4, 27]. Although the EAPC does not give a clear definition of what they mean by existential distress, it can be argued that it is close to that of current literature.

Another poorly investigated issue is that of roles in the management of existential suffering. So far, a few studies have pointed out the importance of different stakeholders such as nurses who are at the front line and can thus better recognize and evaluate existential suffering [19, 28, 29]. Mental healthcare professionals, such as psychotherapists and psychiatrists, and members of religious communities have also been recognized as playing an important role in the management of existential suffering [14].

To our best knowledge, this is the first study investigating the perspectives of professionals on the management of existential suffering when it is part of the request for AS, all the more in a country where AS is legal. This paper, which constitutes the second focus of our study, reports on how Swiss professionals involved in end-of-life care and AS view the management of existential suffering, taking into account the question of roles. This study also explores the difficulties they may encounter along the way and their views on the acceptability of existential suffering as a motive for AS.

## Methods

This exploratory national study involved a full spectrum of persons engaged in end-of-life care and AS in Switzerland. A detailed description of the method was reported in our previous paper [11]. **Table 1** summarizes its main features. A qualitative design was chosen for this study because of the lack of insights on the subject. Sampling was purposive, data were collected from face-to-face interviews, and data saturation was obtained after completing 20 interviews. Elements from the grounded theory approach according to *Corbin and Strauss* [30, 31] were used to develop a conceptual model derived from the data.

In the following results, extracts of the quotations have been selected to illustrate ideas and concepts described by the participants. All transcripts were translated from the original French by the authors. People identification is labelled as follows: Pn (*Participant number*) / PallCP (*Palliative Care Providers*) or PrimCP (*Primary Care Providers*) or EV (*EXIT Volunteer*).

## Results

### Participants

As described in our previous paper [11], twenty-seven professionals were contacted between January and April 2016 to participate in this study. Only one did not answer our request, thus twenty-six people were interviewed. For the second focus of the study presented here, only the stories of 25 participants were included in the analysis due to the accidental shutdown of the recorder during one of the interviews. Participant characteristics are described in *S1 Table*.

### Professionals' perspectives on the management of existential suffering as a motive for requesting assisted suicide

**1. Difficulties when facing these kinds of requests.**   When professionals were asked if they encountered any difficulties when faced with these types of requests, 24 of them reported some kind of struggles. The coding of their statements resulted in two main types of difficulties: 1. Difficulties linked to a feeling of helplessness in the face of these situations and 2.

**Table 1. Design of the study.**

| Qualitative approach | Description |
|---|---|
| *Research paradigm* | Constructivism |
| *Ethical Issues* | • This study was exempted from ethics review by the president of the *CCER* (*Commission Cantonale d'Ethique de la Recherche*, Swiss Cantonal Ethics Committee) because the study entailed no more than minimal risks and was outside the scope of the Swiss Federal Act (Human Research Act, HRA) on research with human participants.<br>• The participation was voluntary and did not involve the collection of personally identifiable information. |
| *Sampling and recruitment strategy* | • A purposive sampling strategy, and more specifically a snowball technique, was used to include participants that differed from background, age and years of experience to ensure different opinions.<br>• A range of persons involved in end-of-life care and assisted suicide were therefore contacted and included into three groups (Palliative Care Providers, Primary Care Providers or EXIT Volunteers).<br>• During this first contact, explanations regarding the goals and the modalities of the study were provided. Participants were also asked if they felt concerned by this issue and if they consented to take part of the study.<br>• No further selection criteria was taken into account.<br>• Written informed consent was obtained before starting the interview. |
| *Sampling saturation* | • Data saturation was obtained after completing 20 interviews<br>• Six additional interviews were conducted without new concepts emerging, to ensure full data saturation. |
| *Data collection period* | Between February and April 2016 |
| *Data collection methods* | • After this first contact, data was collected through face-to-face interviews at a location chosen by the participant.<br>• At the beginning of each discussion the participants were asked their age, gender, job, and how many years of experience they had.<br>• A semi-structured interview guide was used to conduct the conversation *(S2 Table)*<br>• All discussions were tape-recorded, transcribed verbatim and anonymized. |
| *Data analysis* | • February to August 2021 (The analysis was put on hold and resumed 1.5 years after the first study article was published in May 2019) |
| *Units of study* | A story/case reported by participants |
| *Duration of interviews* | 20 to 50 minutes. |
| *Data analysis* | Elements from the evolved grounded theory approach (Strauss & Corbin, 1990) [30, 31] were used to develop a conceptual model derived from the data. As for the first focus of our study, data analysis was based on two steps: open coding and axial coding. No selective coding was applied.<br>• Open coding was used to examine data and conceptualize it into codes. Examples of codes extracted from the quotations of participants talking about the management of existential suffering: explore the suffering, listen to the patient or seeking help from a psychologist.<br>• Axial coding was the second step of the analysis. It consisted of reassembling our codes into broader categories in order to better understand the facets and associations between them. Examples of them: being present or working together. |
| *Criteria to ensure validity* | • Four full recordings were independently coded by three researchers and discussed together at different stages during the analysis.<br>• The coding tree and the development of the analysis was discussed with all researchers in regular meetings.<br>• Quotes per main category were selected by the first author and approved by all co-authors. |

Difficulties related to the conflict of values that these situations may cause. **Table 2** summarizes these findings and include quotes from participants. In general, most of them reported that these requests were challenging and that this could put them in difficulty. We found that, although most participants agreed that existential suffering is no less important than physical suffering, it remains somehow difficult to grasp and very controversial.

**Table 2. Difficulties encountered when facing existential suffering in AS requests.**

| Categories | n* | Codes | n* | Participants' Quotes |
|---|---|---|---|---|
| **Difficulties linked to a feeling of helplessness** | *20* | Feeling of helplessness | 9 | *"I think we also have the right to admit our helplessness. I think it's also healthy to be able to do so. And the idea is not to spread false hopes, that's it. I think we can also affirm our own personal limits; I think that's important. Now it's true that there are many situations, beyond these, there are many situations that put me in front of my own limits, that generate a certain frustration in me, in my possibilities of being able to help the person, in spite of the tools that I can have at my disposal." (P8/PallCP)* |
| | | Not being able to give meaning to the patient's life | 6 | *"How do you expect to be able to give a reason to live to someone who has no one left, who is all alone, who has no family, no friends, and who stays at home staring into space because he can't do anything anymore? (...) Frankly, I don't see how, if we were fairies perhaps..." (P19/EV)* |
| | | Difficult to distinguish between existential suffering and depression | 5 | *"What's difficult for us is also depression, it is all very well to label it as depression, to treat it, but I think the line between the two is not easy either. I understand that a depressed patient who hit rock bottom doesn't want to live anymore but at the same time, when we try to improve it, we don't always manage to give the patient meaning again." (P7/PallCP)* |
| | | Difficult because these are always very complex situations (with no right or wrong answers) | 4 | |
| | | Difficult because I'm not trained to deal with it | 3 | |
| | | Difficult to measure | 3 | |
| | | Difficult to treat | 2 | |
| | | Lack of financial means | 1 | |
| **Difficulties related to a conflict of values** | 10 | Conflict of values | 6 | *"Yes, I am experiencing difficulties that I still find hard to qualify. Finally, I am groping around a bit because I find it strange to live in a society where, for so long, people had to fight for life, to be alive and to live, and now we are fighting to die. I say to myself, "But what kind of society are we in?" I also have difficulties, because in fact my own mother talked to me a lot about EXIT (...) So I am personally very much in conflict with all that." (P5/PallCP)* |
| | | It challenges me | 6 | |

**n***, number of participants who mentioned the category or code (N = 25)

In spite of the above-described difficulties, 7 out of the 25 participants (of which 4 EXIT volunteers) said they were comfortable with these requests, even more so as we get older. Here is an illustration:

*"The older I get, the more I am understanding I am of these requests. That's what Mrs. X used to say*: "*She (another EXIT volunteer) is young, she believes that life is eternal. You're old, you know you're slowly going down so you're aware of that.*"*" (P20/EV)

**2. Acceptability of existential suffering as a motive for AS.** Since the question in our interview concerned the acceptability of existential suffering in requesting AS, not in obtaining it, all participants considered the request "obviously acceptable". The main reason addressed was the fact that one cannot judge the suffering of others, because it is something so personal, intimate and because suffering cannot be limited to a disease status or a somatic cause. In their narrative, they then elaborated on the issue of accessing to this request, and here opinions were more divergent. We found that 10 of the 25 participants could conceive of accessing the request in cases of existential suffering as a last resort, 10 others were against it and 5 participants were undecided and could not say whether they found it acceptable or not. Here is a somewhat sarcastic illustration of a primary care provider who can understand the request but does not find it acceptable to grant it:

"*I think the patient's request is legitimate. Totally legitimate in all cases. Now accessing it is another story. It's really not the same thing. (. . .) It is always legitimate because that's what they want and they have the right to want something because they are in a situation where they think this (AS) is necessary. That being said, in life there are a lot of things that we want but can't have. And then the medical profession has to decide whether they want to become murderers or not.*" (P24/PrimCP)

Among the reasons given for "not acceptable" were participants who seemed generally against AS (for whatever reason), those who said that there must be "more than existential suffering alone" and finally those who said that there are always "other solutions to palliate existential suffering".

In contrast, here is the testimony of a chaplain who believes that AS can be one of the solutions to be used as a last resort, after having exhausted all alternatives, because of the intolerable and refractory nature of this suffering:

"*Yes, this is a fair reason for such a request. Well, she can of course ask because who am I to say she can't? Then you have to question this request and then try to find out why, how and where it comes from, on what it is based. But the request is of course legitimate. Suicide is when the suffering becomes too great. All means are good as long as I can stop this suffering. And when I, as a professional, try a whole panoply of solutions that no longer work or that don't work, well, that's it, I'm going to look for others. And this (AS) is one of the solutions, of course.* (P3/PallCP)

Finally, 5 participants (out of 25) found it difficult to take a position on this issue, such as illustrated by this participant:

"*To say what is right, not right, I can't tell you. I'm a poor human being trying to get by. I think it's a request that we can, that we must understand. If there is a request, we have to hear it.*" (P14/PallCP)

**3. Management of patients with existential suffering.** On the basis of the professionals' accounts, we identified 9 ways of managing existential suffering that were ranked according to the frequency of their occurrence: being present, respect, explore the suffering, give meaning, working together, psychological support, spiritual support, relieve physical symptoms and palliative sedation. A summary of these findings with quotes from participants can be found in **Table 3**.

## Being present

When asked about how existential suffering should be managed, all participants reported the importance of being present with your patient. Presence was seen as a way of creating this particular bond with patients, allowing them to share their human experience, and their suffering. The presence as such was considered to be therapeutic, even if there were sometimes "no other solutions". A few other codes were included in this category, such as including relatives, creating a network or listening, all highlighting how presence can be offered to patients.

## Showing respect

All participants also referred to the importance of showing respect to patients and keeping an open mind. This category refers to an indispensable attitude to have, according to participants,

**Table 3. Management of existential suffering in AS requests.**

| Categories | n* | Codes | n* | Participants' Quotes |
|---|---|---|---|---|
| **Being Present** | **25** | Being present | 20 | *"I attach a certain importance, when we are in such a situation, to the fact that we are, that there is our presence. And I think that maybe facing all this suffering that we can't reduce, well the fact that we can somehow share it, I think it makes sense. (. . .) Why, because we are not alone in the world. Because we exist for each other. And I think that especially in the attitude that I am going to have, as a therapist, to show him that I am interested in what he is going through, maybe I won't be able to respond (to his suffering) (. . .) But the fact that he can say it and share it, I think that makes a lot of sense. I think that an archaic anguish that one can have, that one can live, is to no longer exist for the other, really. To be "alone in the world". And I think that what we can bring as a team, (. . .) is to be able to offer them this presence, this idea that that they exist for us." (P8/PallCP)* |
| | | Including relatives | 15 | |
| | | Listening to the patient | 12 | |
| | | Creating a network around the patient | 10 | *"Even if it's a social suffering with isolated patients, we try to do everything we can. Sometimes it's just putting them in hospital or setting up volunteers, nurses at home or seeing how the family can be reintegrated into the situation. I think all these points are important." (P7/PallCP)* |
| | | Reassuring the patient | 6 | |
| | | Setting up home care volunteers | 6 | |
| | | Setting up home care nurses | 4 | |
| | | Spending time with the patient | 2 | |
| **Showing respect** | **25** | Respecting patients' decision | 21 | *"We tell people what we can do, then we are very respectful of their choice in this house (palliative care unit). That is to say that if someone has already contacted EXIT before, we are not going to try by all means to prevent them from resorting to EXIT. We will explain to them what we can do for them. (. . .) We're just very respectful of their choice, (. . .) we remain very neutral about it. And we say what we can offer according to people's fears and according to what we feel is the reason for their request to EXIT and then the choice is theirs." (P15/PallCP)* |
| | | Not judging patients and keeping an open mind | 6 | |
| | | Showing respect to the patient | 4 | |
| **Seeking to understand the causes of suffering** | **21** | | | *"I often hear myself ask them "But what would make you at some point speed up the process (of assisted suicide)?". And then, depending on what they tell me, it could be pain, it could be the fear of suffocation, it could be the fear of not going through the same thing as her cousin Rose, for example, well that's what I'm exploring. Exploring the fears, exploring the limits of home care, exploring what really motivates (the request)". (P4/PallCP)* |
| **Helping give meaning** | **20** | Helping people regain a sense of identity<br>• Letting them recount their story<br>• Helping them recognize their value (as human beings)<br>• Helping them recognize their role in their family or in society<br>• Enhancing their resources<br>• Helping them regain a sens of identity | 16<br>•<br>13<br>• 7<br>• 5<br>• 5<br>• 4 | *"There is a relevant element. It's that often the illness, the severe illness, irrupted into these people's life and this constituted a threshold. And indeed, with respect to time, there often is a fracture between the "before of who I was once" and the "now" where people often take an « ill person » identity. And their identity is now limited to the issue of being ill. And maybe the idea here is to broaden all this and find out who they were before. So what I'm also doing is 'fishing', looking into their past, to see a little bit who these people were before, what they were doing, and so on. (. . .) What does their illness mean to them in relation to their existence? All of this in the idea of re-establishing a certain continuity between the before and the now of who I am as a sick person." (P8/PallCP)* |
| | | Helping people give meaning to what they are experiencing<br>• Helping them give meaning to what they are experiencing<br>• Helping them maintain hope | 13<br>•<br>13<br>• 4 | |
| | | Suggesting activities | 9 | *"But we still try to add more color to their day, if only for the relationship with the caregivers, by animation activities, we still try to encourage them. (. . .). Let's say that afterwards, it is through the informal daily meetings that we try to reconnect, to make the person realize that life is not so bad, even here (in the retirement home). That it still deserves to be lived for a little while." (P26/PrimCP)* |

*(Continued)*

**Table 3.** (Continued)

| Categories | n* | Codes | n* | Participants' Quotes |
|---|---|---|---|---|
| **Working together** | 18 | Interdisciplinarity | 16 | *"It's a bundle of things that makes people feel better (…) and then you can actually make a small contribution. I think that you have to listen to them, you have to try chemistry (drugs) if it can help, you have to set up the interlocutors they need. Maybe it will rather be a psychotherapist, maybe rather someone for spiritual needs, perhaps someone on the team with whom she really has a great feeling. (…) Then there's a bit of sophrology, then there's relaxation, in short there are all these complementary approaches that we also use to make people feel better." (P15/PallCP)* |
| | | Use of complementary approaches | 6 | |
| | | Having a holistic vision | 4 | |
| | | Calling on the social worker | 3 | |
| **Psychological support** | 15 | Seeking help from a psychologist | 8 | *"Sometimes people are very fearful of the whole "shrink world". And it's true that when you begin offering psychological support, some people think "No, I'm not crazy". I would like to be able to help my colleagues and sometimes myself to succeed in explaining what we do when we deal with existential suffering, what resources we look for and what is the usefulness of a shrink. Whether it is a psychologist or a psychiatrist, they have specialized knowledge." (P4/PallCP)* |
| | | Seeking help from a psychiatrist | 6 | |
| **Spiritual support** | 12 | Exploring spirituality | 8 | *"The religious, spiritual aspect is something that we don't talk about as much. I think we still have to work on that. (…) On the religious aspect, the patients willingly talk about it when it is a question of support, of help. Concerning spirituality, it's a little more complicated. We have patients who talk about it spontaneously, but it's quite rare, and for us, if we want to tackle it, I think we have to have already worked on our own spirituality to know what we're talking about and how we're going to approach it with the patient." (P11/PallCP)* |
| | | Calling on a chaplain | 7 | |
| | | Talking about death (being dead, what happens after) | 3 | |
| | | Talking about religion | 3 | |
| **Relieving physical symptoms** | 11 | Relieving physical symptoms | 9 | *"One of the first things I'm going to look for is the notion of pain, the notion of physical suffering. And besides this, beyond these situations (of requesting AS), I always pay attention to whether the person is in pain. When we are in physical suffering like that, there is often a certain emotion attached to it, a sadness, a loss of hope, and it is not in these conditions that we can reflect. And often the reflex is " I can't take it anymore; I prefer to leave". (P8/PallCP)* |
| | | Use of medication | 5 | |
| | | Calling on an occasional therapist | 2 | |
| **Palliative sedation** | 6 | | | *"And then something that is used a lot here is continuous sedation. It's true that it's more easily used for physical symptoms such as dyspnea, it's something that we propose a lot for dyspnea, but on the other hand we also want to propose it in case of existential suffering, but that's more difficult." (P16/PallCP)* |

**n***, number of participants who mentioned the category or code (N = 25)

when supporting patients with existential suffering. Like the previous category, we can here note the emphasis placed by the participants on "being" rather than on "doing".

### Seeking to understand the causes of suffering

Most of the participants highlighted the importance of taking the time to explore the suffering of these patients, to understand their reasons for requesting AS. According to them, being able to truly understand where the suffering comes from was one of the essential first steps before being able to think about possible solutions. This involving, according to the participants, spending time with the patients, listening to their story, their complaints, their values and wishes for themselves.

### Helping give meaning

20 out of the 25 participants reported the importance of helping people give meaning, accompanying them in this search for meaning, whether it concerns their past, their identity or their current suffering. This actually echoes their interpretations of existential suffering, as described in our previous paper: "a life that isn't worth living any longer" and/or "a life that doesn't make sense anymore". According to these professionals, there are several ways to help them in this regard. The codes grouped in this category are: helping them regain a sense of identity, helping them give meaning to what they are experiencing, and suggesting activities. Each of those includes sub-codes that are listed in Table 3. For many participants, one of the crucial aspects in this idea of helping people "give or regain meaning" was to take care of their sense of identity that may have been damaged by the events of life. While not referring to them by that name, these testimonials were very reminiscent of narrative approaches that are used in existential psychotherapies.

### Working together

The necessity of working together, of using "interdisciplinarity" was mentioned by more than two-thirds of the participants. It should be noted that this term of interdisciplinarity seemed sometimes vague. On the other hand, what was always emphasized was that it is never one person who holds the solution but that responses to existential suffering are plural and therefore emanate from many different people who have to collaborate.

### Psychological support

When confronted to existential suffering, seeking help from a psychologist or a psychiatrist was suggested by many participants and this in a different way from the more vague concept of interdisciplinarity. According to the participants, mental health experts may have "better tools" to explore and to manage this type of suffering.

It should be noted that some participants thought the opposite: psychologists and psychiatrist would not be of great utility when managing existential suffering, as expressed by this participant:

> "*I think the psychologist is useless. Because what are these people going to believe? They will believe that we consider them sick people while I don't think it's a disease.*" *(P21/EV)*

This reluctance to recommend a psychological evaluation because of a possible stigmatization of a psychiatric disorder was, however, found in only three of the participants, thus not reflecting a prevalent phenomenon.

### Spiritual support

Half of the participants mentioned the usefulness of addressing people's spiritual needs. The codes grouped in this category are: exploring spirituality, calling on a chaplain, talking about death (being dead, what happens after) and talking about religion. Their testimonials were not always clear on how to explore spirituality and many reported that it was not always easy to talk about it: because they didn't have the skills, because it wasn't "their job", or because it was such an intimate subject and therefore difficult to discuss with patients.

### Relieving physical symptoms

As expressed by this participant working as a psychologist in palliative care (see **Table 3**), 11 participants recalled the need to, first and foremost, properly manage physical suffering. Relieving physical symptoms was always seen as something we must have thought of before moving on to the next step.

### Palliative sedation

A few participants working in palliative care mentioned the use of continuous sedation to alleviate existential suffering. As the quote in Table **3** illustrates it, most said they were more comfortable when it was applied to the relief of physical symptoms but that this could happen, as a last resort and after having explored alternatives, in case of existential suffering, though recognizing that it was controversial.

**4. Whose role is it to manage existential suffering?.**   When participants were questioned about the roles around managing existential suffering, two types of responses emerged: 14 out of 25 thought that it is a shared responsibility among all health care professionals, pointing out the importance of interdisciplinarity. And among all of them, the most cited were, in order, physicians, psychologists, nurses and chaplains. This is illustrated by the testimony of a palliative care caregiver:

> "*Oh yes, I think that's our role. It's the role of an interdisciplinary team, well we're all interested in it, it's the role of all of us. And when we do interdisciplinary meetings, we try to put things in place at all levels, whether it's at the level of the psychologist, the physician, the caregiver, the occupational therapist, the physiotherapist who can also suggest solutions, the social worker. There you have it, all the professionals revolving around the patient.*" (P1/PallCP)

This part of interview also allowed most of the participants to recall the importance of including patients' relatives. Finally, among participants who thought existential suffering implies a shared responsibility, three of them even suggested that it was the responsibility of the entire society to better take care of these patients.

However, the rest of the participants (11 out of 25) expressed that this task, of managing existential suffering, was above all the prerogative of physicians, especially of general practitioners. We should here clarify that this task was seen more as that of a "conductor" who has to "coordinate" the care/management rather than do everything on his own.

As expressed here, it should be noted that among the six EXIT volunteers, only two considered that it was also their responsibility to manage existential suffering. The rest of the them thought it was not.

## Discussion

Our results showed that almost all participants encountered difficulties when facing existential suffering, most of them related to a feeling of helplessness but also difficulties linked to the conflict of values that these situations can lead to. Regarding the acceptability of existential suffering in the context of an AS request, we found divergent opinions: equally, with one part of the participants thinking that AS was conceivable in cases of existential suffering, when considered as a last resort, while the other part thought the opposite. Five participants were undecided on this issue. Concerning its management, participants expressed several key points and approaches to adopt when taking care of these patients. These include the notion of being present, showing respect, seeking to understand the causes of suffering, helping give meaning, working together, psychological support, spiritual support, relieving physical symptoms and

palliative sedation. As to the question of roles, the opinions of participants differed, with some thinking that responsibility should be shared among all health care professionals, and even at the level of relatives and society as a whole, and others thinking that it is primarily the prerogative of physicians.

Having already pointed out that professionals perceive existential suffering as a composite suffering, it is not surprising that many of them considered it as an acceptable reason for requesting AS. For the same reason, they believed that the management of existential suffering has to be plural and involve an effective interdisciplinary care team. These results corroborate what previous authors suggested when aiming to relieve existential suffering [18, 20, 21]. This also fits with the organization of palliative care units in Switzerland, where interprofessionality is a concept and a method of operation that is now mostly well implemented.

As first steps in dealing with existential suffering, participants focused on crucial attitudes, rather than concrete approaches. Those include the notions of being present for patients and showing them respect, qualities that are not necessarily taught in depth in medical schools. As Amonoo and colleagues [20] point out, "one of the privileges of being a clinician is being present for many of their patients' significant moments–birth, death, life-altering illness, major accidents–but clinicians usually do not receive formal training to address the existential weight of these situations and it is not a clearly defined part of their role." In this regard, several participants suggested that it was necessary to improve the training of physicians, to give them the tools to better listen and support patients with existential suffering. The hidden potentials behind a therapeutic presence have been well described by Covington [28] who defined "caring presence" as a "way of being—of deeply connecting—with another in a relationship" thus providing a "safe space for the patient to share suffering and find meaning in the illness experience".

Also of interest, we found that all participants referred to themes related to existential psychotherapies, like meaning-centered approach or dignity psychotherapy, although they did not refer to them by this label. Those themes included the importance of helping people give meaning and helping them regain a sense of identity. Indeed, as the philosopher and bioethicist Hilde Lindemann expresses it in her book "*Holding and letting go*", this work on identity seems vital: "Serious injury or illness, rape, assault, the death or divorce of a spouse, and other traumas can and frequently play havoc with one's identity. (. . .) All of this contributes to a disintegration of your self. The physician Eric Cassel conceptualizes the sense of this disintegration as *suffering*: to suffer is to feel yourself coming undone. (. . .) It's when we suffer in Cassell's sense of the word that we most need the help of others to hold us in our identities". This idea of the therapist helping to hold people in their identity is in fact central in existential psychotherapies. These approaches, drawing on the philosophy of existentialism pictured in particular by Victor Frankl ("Man's search for meaning") and Irvin Yalom ("Existential psychotherapy"), represent a means of alleviating existential suffering by bolstering meaningful reflection at the end of life [10, 17]. Unfortunately, most clinicians lack training in such approaches and this is certainly a path for improvement in the future. Amonoo and colleagues [20] also proposed a possible framework, utilizing existential themes, and including steps such as "take a narrative history to enhance perspective", "help the patient actively enlist sources of resilience" and "help patients live in accordance with their core values".

Only twelve out the twenty-five participants mentioned the value of exploring patients' spiritual needs when facing existential suffering. This finding may seem rather low taking into consideration that this suffering is sometimes described as a spiritual one. We would argue that spiritual suffering is indeed a major part of existential suffering, but that "not all existential suffering is spiritual suffering" [7]. Although the spiritual dimension has been part of the definition of the concept of health according to the WHO since 2005 [32], the definition of

spirituality has remained somewhat vague until recently, perhaps explaining the reluctance of participants to mention it as such. Interestingly, a 2021 review, looking at the definition of spirituality, found 24 dimensions of spirituality, most often related to "connectedness" (whether to a "higher power", to other people, to one self or even to nature) and "meaning in life" [33]. So perhaps while few participants mentioned exploring spirituality per se, the vast majority did mention aspects considered as components of it in the literature, such as the notion of meaning and therefore providing these people with "spiritual support". That only a few brought up the need for a chaplain could reflect beliefs that this exploration of spirituality is within the reach of everyone (and not only of "spiritual experts"). At this point, these are only hypotheses that would be worth exploring in future studies, as spirituality remains insufficiently addressed in end-of-life care according to recent studies [22, 34–36].

Regarding the result of "palliative sedation" as an alternative to AS in cases of refractory existential suffering, there are also divergent opinions in the literature [4, 13, 14, 37]. A 2020 review [13] reports that "physicians do not hold clear views or agree if and when palliative sedation for existential suffering is appropriate" and that clinicians continue to be more favorable to palliative sedation for physical pain than for existential suffering. Ultimately, whether or not we are open to palliative sedation and/or AS, it is our duty as clinicians, as stated by the SAMS (Swiss Academy of Medical Sciences) to have explored the suffering and to have proposed alternatives. Where to stop in the exploration of alternatives is however still an unanswered question that should be explored in future studies.

Finally, regarding the question of roles, our study did not explore this issue in detail, but it did highlight the fact that all of us, as health care professionals, are concerned by this issue and that we all have a role to play in dealing with existential suffering. This surely includes physicians, who, according to almost half of the participants have an even greater role to play in the management of these patients. Unfortunately, studies on this topic show that patients are dissatisfied with the lack of concern of physicians and nurses for their existential needs [38, 39]. This further reinforces the idea that clinicians' education has to address these dimensions of care. Moreover, the fact that many participants mentioned the role of loved ones, as well as society in general, shows that this issue goes beyond health care professionals and should concern a much wider public in our country.

As this is the first study investigating the perspectives of professionals on existential suffering, its acceptability, and its management in a country where AS is legal, it offers a unique opportunity to reflect on what are desirable responses to existential suffering when it is part of these requests. Existential suffering is plural and its management certainly implies a multiplicity of responses as well. Among these responses are the notions of being present, helping give meaning, providing psychological and spiritual support, skills that are not, in modern medicine, necessarily assigned to physicians. As Kissane [21] points out, "the skills referred to herein are based on the physician as healer, listener, and doctor to the person rather than the symptom or disease".

We hope that our findings will reinforce the work already underway to improve the training of clinicians in the exploration and treatment of existential suffering, as well as to perhaps more systematically implement psycho-existential approaches and spiritual care when confronted with these situations. All questions are far from being answered though and future research should focus on developing and implementing these approaches as well as to reflect on when it can be considered that the exploration of alternatives has been sufficient, implying that AS can become an acceptable alternative when facing existential suffering.

This study has several limitations that have been described in our previous paper. Those include limitations related to the qualitative nature of the study and the fact that the sample size was small and only from the French-speaking part of Switzerland, leading to a lack of

generalization of our results. This study also has the disadvantage of only focusing on the representations of professionals, and not on those of people requesting AS. However, we found that the solutions proposed here are close to what can be found in the previous literature as well as to the few studies that have reported patients' wishes [5, 18–21]. Another possible limitation of this present analysis is that participant were interviewed in 2016 (see timing details in **Table 1**). Since then, in Switzerland, the SAMS adopted in 2018 a broadening of the conditions for access to AS. Indeed, while before 2018, the candidate had to be at the end of life, this criterion was abandoned for the broader one of a suffering deemed "intolerable" by the patient and "comprehensible" by the physician. However, this forthcoming change was already known to Swiss health care professionals as it had already been discussed for several years. We do not believe that the results would be noticeably different today because the question of how to deal with existential suffering is still unclear and AS remains, for those who consider it in cases of existential suffering, a solution of last resort, thus implying the search for prior alternatives.

## Conclusion

This study brings to light that the management of existential suffering has to be plural and involve an effective interdisciplinary team. The approaches proposed here include the notion of being present, showing respect, seeking to understand the causes of suffering, helping give meaning, working together, psychological support, spiritual support, relieving physical symptoms and palliative sedation. Our study also highlights that professionals confronted with these situations still encounter many difficulties and that their training needs to address these issues and give them the tool to better take care of patients with existential suffering. Reinforcing interdisciplinary and multiprofessional approaches is surely also key. We hope that this work will reinforce the efforts that are being made in this direction and that it will also stimulate the ongoing reflection on the acceptability of existential suffering as a reason for requesting AS.

## Supporting information

**S1 Table. Number of participants and their demographics.**
(DOCX)

**S2 Table. Interview guide.**
(DOCX)

## Acknowledgments

The authors wish to show their gratitude to the participants in the study for their time and willingness to be interviewed. They would also like to thank their colleagues at the Institute of Ethics who provided insight on several occasions, as well as Claudia Ricci for reviewing the initial manuscript.

## Author Contributions

**Conceptualization:** Marie-Estelle Gaignard, Samia Hurst.

**Formal analysis:** Marie-Estelle Gaignard.

**Investigation:** Marie-Estelle Gaignard.

**Methodology:** Marie-Estelle Gaignard, Samia Hurst.

**Project administration:** Samia Hurst.

**Resources:** Samia Hurst.

**Supervision:** Sophie Pautex, Samia Hurst.

**Validation:** Sophie Pautex, Samia Hurst.

**Writing – original draft:** Marie-Estelle Gaignard.

**Writing – review & editing:** Sophie Pautex, Samia Hurst.

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
