## [Decision Letter · Decision Letter 0]

9 Aug 2022

PONE-D-22-01201

Existential suffering as a motive for assisted suicide: difficulties, acceptability, management and roles from the perspectives of Swiss professionals

PLOS ONE

Dear Dr. Gaignard,

Thank you for submitting your manuscript to PLOS ONE. After careful consideration, we feel that it has merit but does not fully meet PLOS ONE’s publication criteria as it currently stands. Therefore, we invite you to submit a revised version of the manuscript that addresses the points raised during the review process.

Please pay particular attention to the reviewers' comments regarding the clarity of the concepts used in the results and conclusions, as well as responding to the other concerns they have raised.

We look forward to receiving your revised manuscript.

Kind regards,

Jamie Males

Editorial Office

PLOS ONE

https://journals.plos.org/plosone/s/file?id=ba62/PLOSOne_formatting_sample_title_authors_affiliations.pdf".

Reviewers' comments:

Reviewer's Responses to Questions

**Comments to the Author**

1. Is the manuscript technically sound, and do the data support the conclusions?

Reviewer #1: Yes

Reviewer #2: Yes

Reviewer #3: Partly

2. Has the statistical analysis been performed appropriately and rigorously? 

Reviewer #1: Yes

Reviewer #2: N/A

Reviewer #3: N/A

3. Have the authors made all data underlying the findings in their manuscript fully available?

Reviewer #1: Yes

Reviewer #2: Yes

Reviewer #3: Yes

4. Is the manuscript presented in an intelligible fashion and written in standard English?

Reviewer #1: Yes

Reviewer #2: Yes

Reviewer #3: Yes

5. Review Comments to the Author

Reviewer #1: This paper refers to interviews with health professionals and EXIT volunteers conducted in 2016. Results based on these interviews had been published earlier (Gaignard & Hurst BMC Medical Ethics 2019).

The research question is relevant and the analysis of the data was carefully conducted. The methodology and results are comprehensibly described. The discussion is informative and balanced.

Minor points:

I was a little confused regarding the number of participants: The abstract mentions 26 participants. -apparently only 25 interviews were analysed. The exclusion of one participant was apparently due to a technical problem with the recording (l. 128). Does the "second part of the analysis" (l.. 130) mean the dataset of this paper as a whole? Please clarify this in the abstract and in the results section.

The literature included in the discussion could possibly benefit from an update including more recent publications, e.g. Pesut et al. BMC Palliat Care. 2021, Simmons et al. J Pain Symptom Manage 2022

Reviewer #2: Summary: The manuscript addresses the topic of existential suffering within the context of assisted suicide (AS) in Switzerland. The author engages within recent debates on existential suffering by analysing the ways in which professionals can deal with existential suffering of patients. The first objective of the research is to gain understanding regarding the difficulties professionals can encounter in managing existential suffering. Secondly, the article addresses the question of roles. Thirdly, the article focusses on conflicts that can arise regarding the question whether or not existential suffering is a justifiable reason for AS. The qualitative research relies on twenty-six interviews and uses a grounded theory approach. The scope of the data, as well as the selection of participants of various backgrounds, contributes to the strength of the research and ensures inclusivity of different opinions and attitudes regarding the research topic. The article reads pleasantly and treats a relevant topic.

Methods and theoretical framework: The method of the research is sound and reliable and suits the objectives of the research. The data sample of twenty-six interviews as well as the independent coding of four transcripts guarantees validity. Furthermore, the selected participants seem well-suited for the objectives of the research. Regarding the theoretical framework presented in the background (50-104), a few concepts which seem important to understand and interpret the results are missing. This will be explained in the next section.

Results and conclusion: Overall, the results are presented clearly. However, as mentioned, a few concepts presented in the results are not defined clearly in the introduction. This conceptual unclarity becomes visible in the result section in which the author relates the code ‘talk about death’ to the category of ‘spiritual support’ (table 3). What is meant with spirituality as well as with ‘talk about death’ does not become clear. Depending on what is meant with the concepts, a conversation about death does not necessarily equals a spiritual conversation since a conversation about death could encompass several dimensions. For example, talking about death could refer to (1) the process of dying; (2) the experience of being death; or (3) believes about a possible or non-existent life after death. The first dimension, talking about the process of dying, could also be more of a medical conversation whereas dimension 2 and 3 could be spiritual. Secondly, and this suggestion relies on the above mentioned conceptual unclarity, a definition of spirituality is lacking. The author notes in the discussion that existential suffering sometimes is described as spiritual suffering, yet it does not become clear how the author views spirituality herself in relation to existential suffering (359-61). Since the author proposes spiritual support in the conclusion as one of the desired strategies for dealing with existential suffering, a conceptualization of spirituality and spiritual support seems necessary.

Recommendation: Overall, the author performed an excellent task in clarifying the difficulties professionals encounter when dealing with existential suffering in the context of AS. The manuscript offers interesting and relevant insights as well as useful practical guidelines. If the author elaborates on the definition of spirituality and what is meant with spiritual support, the article could meet the criteria for publication.

Reviewer #3: The paper builds upon an earlier publication from the same qualitative study (Gaignard, Hurst 2019) where the professionals' understandings of existential suffering as part of a wish for assisted suicide have been explored. The main contribution of the present paper are two types of difficulties encountered by these professionals when facing existential suffering in assisted suicide requests and 9 suggested ways of managing existential suffering in such situations. Data are collected in the French speaking part of Switzerland, a country where assisted suicide is legal and there are no legal restrictions in accepting existential suffering as a reason for an assisted suicide request. This is an important qualitative study, and this second part is worth to be published.

The manuscript as it is now however as some serious and other minor flaws that should and certainly could be mended.

- Line 60-63: explain from your earlier paper how professionals called the things that you subsume under 'existential suffering'? Was ist part of actor language? Explain in more detail which "different and sometimes compounded, losses of the dimensions of life" have been encountered, as far as this information is needed to understand the present analysis.

- Discuss and clarify the question, whether existential suffering must be intolerable for the professionals in order to be an acceptable reason. Or did they mean that if existential suffering is among the reasons for requesting assisted suicide it is automatically intolerable?

- Line 85: "our" hospitals are Swiss hospitals?

- Line 91: how does "existential distress" in the EAPC guideline relate to your (or the professionals') definition of existential suffering?

- Line 99ff: Say more clearly that you made a study, which has led to two publications. This paper does not represent the second part of your study but reports from the study with a special focus. It is the second paper from one study! This is unclear for the readers.

- When you speak of "roles" (throughout the paper) it is unclear what you mean. I have the impression that you only mean the professionals' roles, not the roles of friends or family members, who however are certainly important in coping with existential suffering and assisted suicide requests. Be more explicit.

- Table 1: Explain why data collected in 2016 are published only now. In the discussion you should also refer to this fact and explain what has changed in the meantime in Switzerland (if any).

- Table 1: Be more specific about what kind of grounded theory approach you have used. Constructivist? Charmaz? IPA? Expand on the data analysis methodology.

- Line 128-130: You don't need to mention the interview that you could not use for analysis just because it was not recorded. Seems irrelevant.

- Acceptability section: Interesting that you find physicians differentiate between existential suffering as part of a request and the question whether they should act upon the request. Expand on this distinction, since the participants obviously rely on it. I had however the impression that you lump them together under the category of "acceptability". On which level should acceptability be discussed? If you could bring out more details from your data this would be helpful, otherwise it rather obscures a point that participants themselves took seriously. And the information how many participants considered existential suffering to be "a justifiable reason" for requesting remains unclear, if it not clarified what they meant. On one level (line 179) all reasons are just to be accepted out of respect, but on other levels (should they be assisted in suicide? should I assist?) different considerations are important. You could also expand on which reasons have been given by those who said that existential suffering was NOT a justifiable reason for requesting assisted suicide.

- Management section: This is the main part. It should be expanded in several ways. The category "respect" is not completely clear to me: Why is not judging the patient respecting the patient? Is remaining neutral not also a way of being indifferent, and indifference would then not belong to respect? The category "explore the suffering" needs specification, in order to do justice of the good quote that you give in the table, which is more differentiated as your description of the category. The category "give meaning" is not clear to me, should be expanded in more detail. What can be "given" and what will the patient then be capable of doing? Meaning literally cannot be given. Why is the category "psychological support " not included in the category "working together"? Interesting is the dilemma they see when psychological support has the meaning of having a disorder or psychiatric disease, and therefore they hesitate to recommend it.

- Line 333: do you mean interprofessionality, not only interdisciplinarity?

- Line 361 (and also in the description of "spiritual" above: It may be a matter of definition of "spirituality" and "spiritual needs", when, as you say, fewer than expected interviews mentioned them. What was your definition? Are there definitions given by participants or to be derived from the interviews?

6. PLOS authors have the option to publish the peer review history of their article (what does this mean?). If published, this will include your full peer review and any attached files.

Reviewer #1: No

Reviewer #2: No

Reviewer #3: **Yes: **Christoph Rehmann-Sutter

---

## [Author Response · Author response to Decision Letter 0]

8 Jan 2023

Please see the attached document entitled "response to reviewers".

---

## [Decision Letter · Decision Letter 1]

20 Feb 2023

PONE-D-22-01201R1Existential suffering as a motive for assisted suicide: difficulties, acceptability, management and roles from the perspectives of Swiss professionalsPLOS ONE

Dear Dr. Gaignard,

Thank you for submitting your manuscript to PLOS ONE. After careful consideration, we feel that it has merit but does not fully meet PLOS ONE’s publication criteria as it currently stands. Therefore, we invite you to submit a revised version of the manuscript that addresses the points raised during the review process.

 As you will see from the comments below, two of the reviewers are happy with the revisions to your manuscript. However, one reviewer has a request for clarification. Please could you revise your manuscript to address this request?

We look forward to receiving your revised manuscript.

Kind regards,

Steve Zimmerman, PhD

Associate Editor, PLOS ONE

Journal Requirements:

Reviewers' comments:

Reviewer's Responses to Questions

**Comments to the Author**

1. If the authors have adequately addressed your comments raised in a previous round of review and you feel that this manuscript is now acceptable for publication, you may indicate that here to bypass the “Comments to the Author” section, enter your conflict of interest statement in the “Confidential to Editor” section, and submit your "Accept" recommendation.

Reviewer #1: All comments have been addressed

Reviewer #2: All comments have been addressed

Reviewer #3: (No Response)

2. Is the manuscript technically sound, and do the data support the conclusions?

Reviewer #1: Yes

Reviewer #2: Yes

Reviewer #3: Yes

3. Has the statistical analysis been performed appropriately and rigorously? 

Reviewer #1: N/A

Reviewer #2: N/A

Reviewer #3: Yes

4. Have the authors made all data underlying the findings in their manuscript fully available?

Reviewer #1: Yes

Reviewer #2: Yes

Reviewer #3: Yes

5. Is the manuscript presented in an intelligible fashion and written in standard English?

Reviewer #1: Yes

Reviewer #2: Yes

Reviewer #3: Yes

6. Review Comments to the Author

Reviewer #1: I enjoyed reading the revised paper. By carefully adressing the comments (especially from reviewer 3) the categories have gained further clarity . However, the new subcategory " Showing respect and being open to their request" seems ambiguous to me: Does "open" mean "willing to consider the request (to hasten death)” - or rather something along the lines of "feeling empathy"?

l. 170: Acceptability of existential suffering as a motive for AS: It has now become clear that and how a distinction was made between the acceptability of the request for AS and the acceptability to grant AS in this case. However, it is now not entirely clear whether the interviewees meant rather the comprehensibility of the request because of an unbearable suffering (in the sense of a distressing symptom) or the acceptability of the desire (in moral terms). In other words: there is a difference between “understanding that someone has a certain wish” and “finding it morally acceptable that someone has a certain wish”.

Reviewer #2: (No Response)

Reviewer #3: Very convincing and carefully written.

7. PLOS authors have the option to publish the peer review history of their article (https://journals.plos.org/plosone/s/editorial-and-peer-review-process#loc-peer-review-history). If published, this will include your full peer review and any attached files. Do you want your identity to be public for this peer review? (Answer options: Yes, No)

If the authors want to publish it, I am fine with it.

7. PLOS authors have the option to publish the peer review history of their article (what does this mean?). If published, this will include your full peer review and any attached files.

Reviewer #1: No

Reviewer #2: No

Reviewer #3: **Yes: **Christoph Rehmann-Sutter

---

## [Author Response · Author response to Decision Letter 1]

30 Mar 2023

Please see the "Reponse to reviewwers" document in the attached files

---

## [Editor Report · Decision Letter 2]

6 Apr 2023

Existential suffering as a motive for assisted suicide: difficulties, acceptability, management and roles from the perspectives of Swiss professionals

PONE-D-22-01201R2

Dear Dr. Gaignard,

We’re pleased to inform you that your manuscript has been judged scientifically suitable for publication and will be formally accepted for publication once it meets all outstanding technical requirements.

Kind regards,

Vanessa Carels

Staff Editor

PLOS ONE
---

## [Editor Report · Acceptance letter]

13 Apr 2023

PONE-D-22-01201R2 

Existential suffering as a motive for assisted suicide: difficulties, acceptability, management and roles from the perspectives of Swiss professionals 

Dear Dr. Gaignard:

I'm pleased to inform you that your manuscript has been deemed suitable for publication in PLOS ONE. Congratulations! Your manuscript is now with our production department. 

Kind regards, 

on behalf of

Dr. Vanessa Carels 

Staff Editor

PLOS ONE